# Generative Pre-Trained Transformer-Empowered Healthcare Conversations: Current Trends, Challenges, and Future Directions in Large Language Model-Enabled Medical Chatbots

**James C. L. Chow** [1,2,*], **Valerie Wong** [3] and **Kay Li** [4]

1   Department of Medical Physics, Princess Margaret Cancer Centre, University Health Network, Toronto, ON M5G 1X6, Canada
2   Department of Radiation Oncology, University of Toronto, Toronto, ON M5T 1P5, Canada
3   Department of Physics, Toronto Metropolitan University, Toronto, ON M5B 2K3, Canada; valerie.wong@torontomu.ca
4   Department of English, University of Toronto, Toronto, ON M5R 2M8, Canada
*   Correspondence: james.chow@uhn.ca

**Abstract:** This review explores the transformative integration of artificial intelligence (AI) and healthcare through conversational AI leveraging Natural Language Processing (NLP). Focusing on Large Language Models (LLMs), this paper navigates through various sections, commencing with an overview of AI's significance in healthcare and the role of conversational AI. It delves into fundamental NLP techniques, emphasizing their facilitation of seamless healthcare conversations. Examining the evolution of LLMs within NLP frameworks, the paper discusses key models used in healthcare, exploring their advantages and implementation challenges. Practical applications in healthcare conversations, from patient-centric utilities like diagnosis and treatment suggestions to healthcare provider support systems, are detailed. Ethical and legal considerations, including patient privacy, ethical implications, and regulatory compliance, are addressed. The review concludes by spotlighting current challenges, envisaging future trends, and highlighting the transformative potential of LLMs and NLP in reshaping healthcare interactions.

**Keywords:** AI; humanistic AI; ethical AI; machine learning; large language models; natural language processing; medical chatbot; transformer-based model; ChatGPT; healthcare

## 1. Introduction

The trajectory of artificial intelligence (AI) development spans decades, with machine learning (ML) emerging as a pivotal force in propelling AI's evolution [1–4]. The adoption of AI and ML in the medical field has experienced significant growth, particularly in ML-enabled medical devices. Joshi et al. focused on 691 FDA-approved AI/ML-enabled medical devices, revealing a substantial surge in approvals since 2018, predominantly in radiology. The prevalence of the 510(k) clearance pathway, relying on substantial equivalence, is notable [5]. This review focuses on a specific ML facet: the Large Language Model (LLM) within Natural Language Processing (NLP) [6,7]. Particularly, we delve into the integration of LLMs like Chat Generative Pre-trained Transformer (ChatGPT, version 3–4) into chatbots, augmenting their capacity for seamless user engagement [8–10].

Chatbots, AI-driven conversational agents prevalent in online interactions, have found extensive utility in disseminating healthcare information and enhancing customer services [11–15]. Table 1 summarizes the general features that medical professionals would expect a medial chatbot to have. These features encompass accurate information retrieval, symptom assessment, and diagnosis support to help in understanding and addressing health concerns. Moreover, the chatbot is expected to provide treatment guidance, medication information, and assistance with appointment scheduling, ensuring a comprehensive

healthcare experience. Health monitoring features, emergency response capabilities, and patient education contribute to a holistic approach. Privacy and security measures, multilingual support, and integration with electronic health records uphold standards of confidentiality and accessibility. Personalized recommendations, follow-up mechanisms, and a user-friendly interface tailor the chatbot experience to individual needs, while features like adherence support and mental health resources further enhance its utility. Continuous feedback mechanisms ensure ongoing improvement, making the chatbot a valuable tool to promote patient well-being. The advent of ChatGPT has notably elevated the appeal of chatbots, facilitating more human-like interactions through adaptive text learning [16,17]. However, the precision of healthcare information dispensed by ChatGPT still raises some concerns, prompting inquiries into potential user misguidance [18–20].

**Table 1.** Features of AI chatbot expected by the medical professional.

| Feature | Description |
|---|---|
| Accurate Information Retrieval | Provide accurate and up-to-date medical information from reliable sources. |
| Symptom Assessment | Analyze and assess user-described symptoms to suggest potential health conditions. |
| Diagnosis Support | Offer preliminary assistance in suggesting potential diagnoses, understanding its limitations. |
| Treatment Guidance | Provide general information on treatments, medications, and lifestyle recommendations. |
| Medication Information | Offer details about medications, including dosage, side effects, and potential interactions. |
| Appointment Scheduling | Assist users in scheduling appointments with healthcare providers and send reminders. |
| Health Monitoring | Support users in tracking and monitoring health metrics like blood pressure or blood sugar. |
| Emergency Response | Recognize urgent situations and provide emergency response information or facilitate contacts. |
| Patient Education | Offer educational content to enhance users' understanding of medical conditions and prevention. |
| Privacy and Security | Ensure strict adherence to data privacy regulations and maintain the confidentiality of user health information. |
| Multilingual Support | Provide communication in multiple languages to cater to diverse patient populations. |
| Integration with EHR | Facilitate integration with existing healthcare systems to access relevant patient data. |
| Personalized Recommendations | Offer personalized health advice based on user data, preferences, and lifestyle. |
| Follow-up and Continuity of Care | Implement features for follow-up interactions, reminders, and maintaining continuity of care. |
| User-Friendly Interface | Ensure an intuitive and user-friendly interface for easy interaction. |
| Adherence Support | Assist patients in adhering to prescribed treatment plans and medications. |
| Mental Health Support | Include features for mental health assessments, stress management, and access to mental health resources. |
| Feedback and Improvement | Incorporate mechanisms for users to provide feedback on the chatbot's performance. |

This review underscores the dual role of AI-assisted healthcare chatbots, exploring their potential to educate the public with accurate information sourced from medical institutions [21], while acknowledging the risks associated with misinformation [22]. Furthermore, the exploration extends to uncharted territory, considering the role of chatbots in aiding disabled individuals and the elderly. Additionally, the paper contemplates the nuanced function of chatbots as temporary emotional outlets, particularly relevant given the upsurge in depression cases during the recent pandemic [23,24].

In this comprehensive examination, we scrutinize the advantages that AI chatbots bring to the healthcare system while addressing inherent challenges. The discussion encompasses the intricate dynamics of AI chatbots, their potential to positively impact healthcare information dissemination, and their pitfalls. As we delve into the future trajectory, the review aims to illuminate potential advancements in AI chatbots within the healthcare sector.

## 2. Fundaments and Evolution of Language Models

### 2.1. Fundamentals of Natural Language Processing

NLP stands as a cornerstone in the realm of ML, a subset of AI that learns from data to approximate human expectations [25,26]. Particularly, NLP plays a pivotal role in facilitating AI's comprehension of the diverse languages used by individuals. Chatbots integrated with NLP capabilities excel in learning and understanding the natural language patterns employed by users in textual communication, enabling them to respond intelligibly [27,28]. Figure 1 shows a typical chatbot architecture, including the user interface, user message analysis component, dialog management component, responses generation component, and the database [28]. The NLP is mainly linked to the message analysis component to analyze the context information.

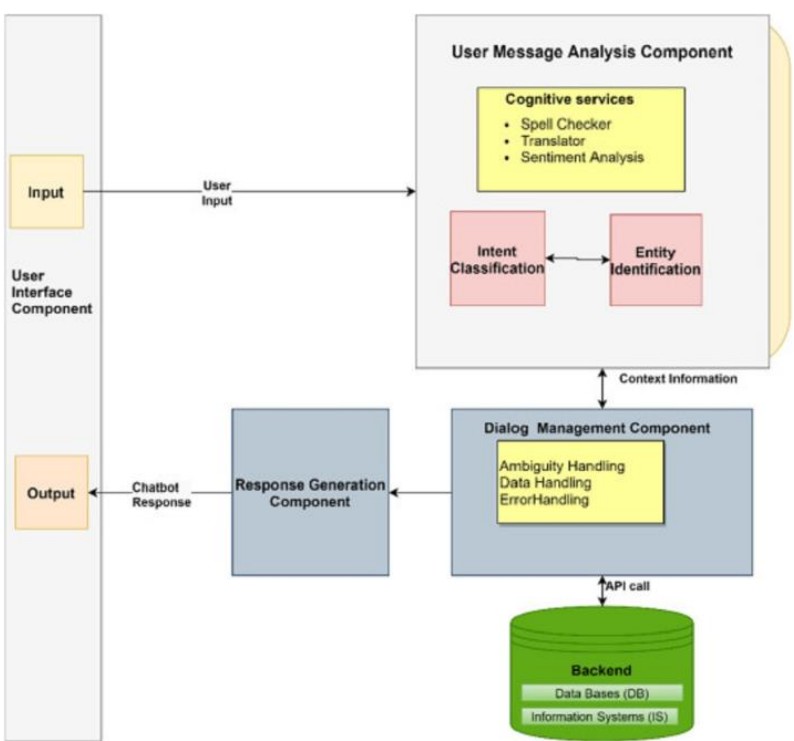

**Figure 1.** General chatbot architecture. Source: Adapted from [28].

Given the inherent variability in how individuals communicate, lacking a standardized template or exact pattern, ML, especially NLP, strives to analyze free-text and speech through linguistic and statistical algorithms. This analysis aims to extract discernible patterns from the rich tapestry of human expression [29,30]. While pattern analysis forms the foundation, the evolution of AI necessitates its ability to engage in meaningful conversations with users, primarily exemplified in question-answering (QA) scenarios [31,32].

The acquired text patterns are cataloged in a database, empowering the AI to match these learned patterns during user interactions—a process akin to pattern matching and text searching techniques [33]. Crucially, NLP goes beyond mere pattern recognition; it grapples with the nuances of how individuals articulate ideas. This involves understanding

that distinct expressions can convey the same meaning, enabling AI to emulate human-like responses, thus enhancing the conversational experience [34,35].

It can be seen that the training of the model involves exposing the algorithm to vast amounts of text data, allowing it to learn the patterns, semantics, and structures inherent in human language. This process typically utilizes large datasets to train the model on tasks such as language understanding, sentiment analysis, or question answering. The NLP model undergoes iterative adjustments during training, refining its ability to recognize and generate meaningful language output. The ultimate goal is to enhance the model's proficiency in understanding and generating human-like text, enabling it to perform diverse linguistic tasks with accuracy and relevance.

In the healthcare domain, NLP demonstrates its prowess by extracting pertinent information from free-text documents such as electronic health records. Beyond symptom examination, NLP's ability to compare, classify, and recommend actions based on vast sets of textual data contributes significantly to disease symptom classification and patient guidance [36,37]. NLP emerges as a linchpin in the intersection of AI and healthcare, fostering a nuanced understanding of language patterns, enhancing conversational dynamics, and contributing invaluable insights in the medical field [38]. The following exploration further delves into the applications and implications of NLP in healthcare conversations.

## 2.2. Evolution of Large Language Models

LLMs have emerged as transformative components within the NLP, significantly influencing the evolution of AI [39,40]. LLMs, belonging to the broader category of machine learning, excel in processing and generating human-like text by leveraging extensive datasets. Their remarkable ability to capture intricate language nuances and generate coherent responses has positioned them as integral players in advancing NLP [41]. In healthcare conversations, LLMs play a crucial role in enhancing the conversational capabilities of AI systems [42]. By understanding and generating contextually relevant responses, LLMs contribute to the humanization of interactions, creating more engaging and effective healthcare dialogues.

Several key LLMs have made a significant impact on the healthcare conversations, revolutionizing the way AI engages with users. One noteworthy exemplar is OpenAI's GPT (Generative Pre-trained Transformer) series, with models like GPT-3 and 4 demonstrating exceptional language understanding and generation capabilities [43,44]. GPT-4, in particular, has garnered attention for its versatility in various applications, including healthcare-related tasks [45]. BERT (Bidirectional Encoder Representations from Transformers) is another influential LLM that has left an indelible mark on NLP [46,47]. Renowned for its bidirectional training approach, BERT excels in grasping contextual nuances, making it particularly adept at understanding the intricacies of medical language and information. Furthermore, models like XLNet [48], T5 (Text-to-Text Transfer Transformer) [49], and BART (Bidirectional and Auto-Regressive Transformers) [50] have played instrumental roles in advancing the sophistication of LLMs in healthcare applications. These models exhibit enhanced capabilities in processing medical literature, extracting relevant information, and generating coherent responses tailored to healthcare-related inquiries [51]. Table 2 shows some popular LLMs used in healthcare conversations.

The utilization of LLMs in healthcare conversations signifies a paradigm shift, enabling AI systems to comprehend and respond to user queries with a depth of understanding akin to human-like interactions. As we navigate through the evolutionary trajectory of LLMs, their continued refinement and integration into healthcare dialogue systems hold promise for further augmenting the efficacy and user experience in the realm of medical conversations.

**Table 2.** Some LLMs and their potential applications in healthcare conversations.

| Model | Description | Applications in Healthcare |
|---|---|---|
| GPT-3 and 4 [43–45] | OpenAI's powerful LLM with strong natural language understanding. | Medical documentation, question answering, text-based interactions |
| BERT, BioBERT and ClinicalBERT [46,47] | Bidirectional processing makes BERT suitable for clinical text analysis. | Clinical text analysis, medical literature understanding, biomedical text mining, information extraction from medical texts, clinical note understanding, medical question answering |
| XLNet [48] | OpenAI's model capable of capturing bidirectional context. | Medical literature analysis, clinical documentation |
| T5 [49] | Text-to-Text Transfer Transformer, designed for various NLP tasks. | Summarization of medical documents, question generation |
| BART [50] | Bidirectional and Auto-Regressive Transformer, used for text generation. | Text summarization, document generation, paraphrasing |

The incorporation of LLMs into medical chatbots introduces significant advantages, revolutionizing the healthcare interactions. LLMs enhance the understanding of contextual nuances in medical queries, enabling chatbots to provide more accurate and relevant responses [52]. This heightened comprehension fosters a humanized interaction, with LLMs proficiently mimicking natural language patterns, creating a more engaging and empathetic user experience. Additionally, LLMs empower medical chatbots to efficiently retrieve and disseminate precise medical information, positioning them as reliable sources of up-to-date healthcare knowledge [53]. Figure 2 shows the touchpoints of a patient's care journey in which an LLM can be employed to enhance the patient's experience.

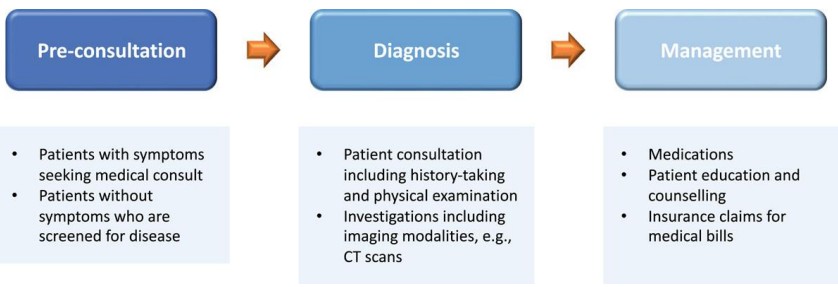

**Figure 2.** A standard patient journey in healthcare, encompassing three key stages: (1) pre-consultation involves patient registration, medical consultation, or health screening; (2) diagnosis includes patient consultations, examinations, and supplementary investigations; and (3) management comprises medication, patient counseling, education, and reimbursement for medical bills. LLMs exhibit potential to improve the patient experience at each touchpoint in this journey. Source: Adapted from [53].

However, the implementation of LLMs in medical chatbots is not without challenges. Ensuring the accuracy and trustworthiness of information is paramount, as LLMs may inadvertently generate inaccurate responses, posing a risk of misinformation [54,55]. Privacy and security concerns arise, demanding robust measures to safeguard sensitive health-related data [56]. Furthermore, interpreting complex medical terminology and aligning with user expectations present ongoing challenges. Addressing these hurdles is essential to fully harness the potential benefits of LLMs in the dynamic realm of medical chatbots [18].

### 3. Results

*3.1. Application of AI Chatbot in Healthcare*

3.1.1. Healthcare Knowledge Transfer with Chatbot

As mentioned, the general public are more likely to use chatbots to look for answers to their questions in daily life or even in medicine. So it is important to note that chatbots might obtain wrong information and mislead users, unless developing a healthcare-based chatbot that is designed to answer people with accurate medical information [57,58]. People would not know if they were misled and believe the provided information blindly, which could lead to accidents. It would be best to develop medical chatbots with professionals like doctors and nurses to ensure the information is accurate and easy to understand for the general public. If LLMs could be trained specifically in healthcare then their trustworthiness would be increased, but this might require professionals to verify that every single piece of information provided by the LLM is correct [59]. LLMs' trustworthiness is controversial; some researchers think they simply gather information from the internet and provide it to the users, while other researchers think they have the potential to be trained specifically for healthcare purposes with related journals [60]. On the other hand, using chatbots for healthcare knowledge transfer can help prevent clinicians from answering similar questions from various patients repeatedly, and hence could allow clinicians to work on jobs that are highly prioritized [61]. As such, there remains a concern of accuracy and reliability of the information that researchers always keep in mind [18].

3.1.2. Symptom Diagnosis

According to the study of Kumar et al. [62], training an AI model with a number of papers in the healthcare field means that the model is able to analyze and predict the symptoms of diseases. Aside from simply classifying symptoms, clinicians are responsible for compiling electronic health records (EHRs), which is a patient information managing digital system, for every single patient visiting a hospital [63]. Having an LLM trained for classifying symptoms according to doctor–patient conversations would increase the efficiency of seeing each patient and thus reduce the chance of overcrowding during busy hours [63]. In fact, there is already a ChatGPT-like chatbot created for healthcare, Med-PaLM, which is able to analyze X-ray images according to the examples given in [64]. Moreover, it is possible for the LLM to integrate with telehealth services especially for those who have a disability [53]. In addition, it is believed that LLMs are even able to detect later-life depression, which is a kind of major public health concern that occurs in the older generation, as the name states [65]. NLP analyzes the way people talk and also the speed and pitch they use in order to understand their speech patterns [65]. Having understood their speech patterns, it is possible to discover the speech patterns of people with depression and use these as a kind of template to analyze and compare the template patterns with the target user in order to determine whether they are possibly experiencing later-life depression.

3.1.3. Mental Healthcare

Having a conversation with someone is the easiest way to help balance mental health, as people are able to express their feelings when they talk to someone. It is not difficult to imagine that people can have conversations with LLM chatbots like ChatGPT especially when there is no one to talk with [66,67]. Stress could simply keep building up when something bad happens in a person's daily life and they cannot talk to someone, and then the stress reaches a limit and their performance at work may worsen and their health might also be affected to some extent. Chatbots can help with daily emotional support; there are studies proving they are capable of helping people get rid of stress and feelings of depression, which also somehow demonstrate better results than traditional mental heath treatments [68]. With NLP analyzing the texts posted on different social media platforms, the AI model is able to perform detection of emotions and monitoring of mental health [69]. It is a sort of text classification scenario in that the NLP technique allows the

AI model to analyze texts and compare them with other similar texts and classify them into different cases in order to detect mental illness from those [70]. In social media, AI models most commonly focus on detecting scenarios like suicide; when they detect any wording that might possibly relate to suicide, it recommends users to contact mentors for mental support [71]. Other than that, there are many people living alone and some of them might not have anyone around them with whom they can share their feelings of daily life, for example, they cannot express the negativity they feel at school or in work. People get depressed when negativity and stress keep building up without letting it out by having someone to talk to; so, it would be nice if chatbots can become a sort of a place for people to let out their stresses [72]. It is hard to live alone in society, especially when there are infinite factors that can make people have a rough day and stress builds and bursts out when there is no way to release it; people can get angry at no one without a reason and this might lead to a fight, which could bring down the quality of life around the community.

### 3.2. Ethical and Legal Implications

### 3.2.1. Patient Privacy and Data Security Concerns

The integration of AI, particularly LLMs, into healthcare conversations brings forth ethical and legal considerations, with the foremost among them being patient privacy and data security. As medical chatbots process sensitive health information, ensuring robust measures for data encryption, storage, and transmission becomes paramount [73]. Ethical considerations demand that patient data are handled with the utmost confidentiality and that stringent protocols are in place to prevent unauthorized access or breaches, safeguarding the trust patients place in AI-assisted healthcare interactions [74,75].

### 3.2.2. Ethical Considerations in AI-Assisted Healthcare Conversations

Beyond privacy concerns, ethical considerations play a pivotal role in the deployment of AI-assisted healthcare conversations. Ensuring transparency and informed consent becomes crucial when patients engage with medical chatbots. The ethical development and use of AI models, including LLMs, involve addressing biases, avoiding discrimination, and maintaining fairness in the provision of healthcare information [76,77]. Striking the right balance between technological advancements and ethical principles is essential to build a foundation of trust between patients, healthcare providers, and AI systems [78].

### 3.2.3. Regulatory Compliance in AI-Powered Healthcare Applications

Effectively managing regulatory complexities is a multifaceted challenge when incorporating AI-powered healthcare applications. It is crucial to uphold adherence to established healthcare regulations, exemplified by the Health Insurance Portability and Accountability Act (HIPAA) in the United States. Ensuring compliance with these regulatory standards is not only essential for safeguarding patient rights but also forms the bedrock for responsible AI deployment [79]. This adherence mitigates legal risks and fosters a seamless integration of technology into healthcare practices. The evolving dynamics of health-related conversations, driven by AI chatbots, necessitate a thorough understanding of ethical and legal implications. As the regulatory landscape continues to evolve, it is noteworthy to mention the European Union's AI Act [80], which introduces regulations specific to AI systems, emphasizing transparency, accountability, and user safety in the deployment of AI technologies across various sectors [81]. Table 3 succinctly outlines the concerns related to AI chatbots, encompassing ethical and legal dimensions. It emphasizes Patient Privacy and Data Security, stressing the need for robust encryption and storage. The Ethical Considerations highlight transparency, informed consent, and fairness, while Regulatory Compliance underscores adherence to regulations like HIPAA and the European Union's AI Act, ensuring a responsible AI deployment framework aligned with legal and ethical standards.

**Table 3.** Concerns of AI-assisted healthcare conversations with ethical and legal implications.

| Concern | Description |
|---------|-------------|
| Patient Privacy and Data Security | AI in healthcare raises concerns about patient data security. Robust measures are needed for encryption and storage. |
| Ethical Considerations in AI-Assisted Healthcare | Transparency and informed consent are crucial. Addressing biases and maintaining fairness in healthcare is essential. |
| Regulatory Compliance in AI-powered Healthcare | Adhering to healthcare regulations like HIPAA and the European Union's AI Act is crucial. It establishes a framework for responsible AI deployment. |

As we delve deeper into the ethical and legal implications, it is evident that maintaining a delicate balance between technological innovation, patient privacy, and regulatory compliance is crucial for the responsible and sustainable evolution of AI-assisted healthcare conversations [18]. Subsequent sections will explore specific strategies, best practices, and ongoing developments aimed at addressing these multifaceted considerations.

## 4. Discussion

The integration of conversational AI into established healthcare systems introduces a spectrum of challenges, with technical hurdles at the forefront. Concerns encompassing compatibility, interoperability with electronic health records, and potential resistance from healthcare professionals pose substantial obstacles to the seamless assimilation of AI into healthcare workflows. Beyond technical considerations, adherence to regulatory standards and healthcare laws emerges as a paramount concern. This section explores the intricate challenges tied to ensuring that AI applications in healthcare align with existing regulations, taking into account regional variations in compliance requirements. Moreover, the risk of over-reliance on AI-driven solutions looms large, raising questions about diminished human oversight and decision making. Striking a delicate balance between leveraging technology for efficiency and preserving the human touch in healthcare becomes imperative to ensure patients receive care that is both personalized and empathetic. In addition, the implementation of advanced AI systems, particularly LLMs, bears significant financial implications. The costs associated with acquiring, implementing, and maintaining these technologies require careful consideration, particularly for healthcare facilities operating with limited resources. These complex challenges emphasize the nuanced nature of incorporating AI into healthcare practices.

### 4.1. Future Development and Challenges

It is believed that AI would evolve much quicker in the industry and especially the chatbot in healthcare. It could be capable of performing symptom diagnosis more precisely or effectively helping with people's mental care [82,83]. However, there are concerns that remain unsolved like privacy problems. On the other hand, there is a chance that not all kinds of individuals understand the use of chatbots or are able to use chatbots, such as people with disabilities and the elderly. Also, there are questions surrounding whether AI chatbots are responsible for legal problems like plagiarism [84]. Regulations and uses of LLMs remain controversial between the general public; people are still be concerned about the above-mentioned aspects, so it is important to make LLMs a convenient and secure tool for people to use comfortably without worrying about anything. Additionally, LLMs should not be misused.

People question the security of using LLMs in healthcare when it comes to privacy concerns, especially applying LLMs in healthcare. As LLMs could be able to handle patients' health records and some other important personal information, then it is questionable whether the programming of LLMs is secure enough for personal data management [85].

Researchers are urged to look for a way to strengthen the protection of patients' sensitive information in order to allow people to use these models comfortably. In addition, the regulation of the use of data and information for LLMs remains unclear and improper, and hence any unauthorized access or misuse of LLMs is not preventable [86]. On the other hand, it is required for the LLM to obtain access to health records so that it is pre-trained for healthcare knowledge specifically to enhance its precision with information regarding medicines [87]. Although it is impossible to prevent LLMs from accessing possibly sensitive information, improving its security in terms of the way it stores or uses that information needs to be focused on. Therefore, it is important to set up regulations for LLMs in different fields, not just healthcare. However, regulations could be very different globally as there are various standards on such controversial topics [85]. It is also believed that transparency of the development of LLMs is important so that the trust from the general public can be maintained and so that people might have the chance to understand what kind of data are necessary and how they would be used for LLM development in order to apply it into healthcare [88]. In addition, it is about ensuring that the actions of the LLMs using those patients' sensitive information for whatever need meet the privacy and security requirements of the HIPAA [89].

There is nothing that can satisfy every single individual in the world, including the seemingly perfect LLMs. Then, there is a question as to whether LLMs can satisfy everyone, even those with disabilities and the elderly. For example, what if people with disabilities or elderly people live alone and there is no one who can tell or teach them how to use LLMs; then, they might never know what it is, how to use it, and even why they need to use it. They might not be able to enjoy the convenience brought by the LLMs. Of course, they do not have to use it to enhance their daily life as they used to live in the usual way, but it is unfair to them that they cannot always enjoy what other people can. On the other hand, they might then exhibit symptoms of later-life depression, unless they receive increased care from their surroundings, such as having people to talk with, play with, or even just having a walk around the street; all of these factors would help them to get rid of the stress from daily life, as people have rough days sometimes and they need a way to let out their emotions. Ideally, if people with disabilities and the elderly can easily learn how to deal with new technologies and use them, then it could possibly make everyone's life easier and less trouble [90]. Because not all people will accept using new technology, even people who are not disabled or elderly, we should not force everyone to learn and use LLMs but information regarding their positive and negative factors should be disseminated to the general public, as well as how developers would improve LLMs in different aspects to ease the concerns that people have about them. As a result, people could gradually increase their acceptance of using such technology. Hence, it is important to have volunteers to help spread healthcare LLMs' advantages and teach them to use them; once everyone learns how to use LLMs correctly, they should be able to explore more functions themselves without help. Then, people have someone to have a conversation with; even when it is late at night and their family or friends are not available, they could still have a chatbot to talk with, get advice from, and comfort their feelings to some extent [91].

The concept of hallucinations in LLMs refers to instances where the model generates outputs that are factually incorrect, misleading, or unrelated to the input context. For medical chatbots, hallucinations can have significant implications. If an LLM produces inaccurate medical information or provides recommendations based on false premises, it can compromise the reliability of the chatbot. This potential misinformation may lead to misunderstandings, misdiagnoses, or inappropriate medical advice, posing risks to users' well-being. Managing and mitigating hallucinations in LLMs is crucial for ensuring the trustworthiness and safety of medical chatbots, underscoring the importance of ongoing refinement and validation processes in their development. About the ethical concerns with academic research, researchers questioned LLMs like ChatGPT about whether the resources it provides to the users for research uses on academic work are reliable or not. According to the research of Guleria et al. [92], ChatGPT was asked to write an article with a specific

topic and to provide the resources it used in the article. The contents it wrote for the article were deemed to be correct; however, the resources it listed as those used for the article could not be found by the researchers themselves. As a result, the researchers believe that ChatGPT is not as reliable as people think because the resources it provided could be made up by the program itself, instead of researching and analyzing the scientific literature from the internet [92]. In addition, the researcher performed this experiment to test ChatGPT's accuracy on the information related to academic research, and it seemed to be not accurate enough in some fields; therefore, it is required to verify the output generated by ChatGPT to prevent any misinterpretation [93]. Moreover, this shows that people should not blindly trust the information provided by ChatGPT, especially information related to medicine and any healthcare knowledge, as it could be putting the health and safety of the general public in danger and also lead to medical misconduct in serious cases, and it cannot even be responsible for any incorrect information it provides in any content of the scientific literature [92]. Moreover, it cannot be proven whether the LLM would provide a similar article to different users if they ask it to write articles on the same topic, such as academic essay assignments; then, it could lead to a large scale of plagiarism. Because the LLMs are trained with scientific studies that are copyrighted, the information or output, like articles it is commanded to provide, could be copied from those scientific journals and people might not know about this, causing plagiarism as a result [94]. Therefore, all LLM-generated texts should be tested with a plagiarism detector. According to the research of Gao et al. [95], AI-generated output is easily found in generated abstracts instead of original abstracts, compared with a median of 99.98% for generated texts and a median of 0.02% for the original texts [95].

On the other hand, researchers are looking forward to seeing improvement in LLMs in the aspect of explainability, as they found that LLMs lack the ability to explain something well with the use of detailed, step-by-step explanations for the information it provides for users [96]. Moreover, it is also believed that the market size of chatbots will have a significant increase in the future, as they keep improving and evolving [97].

The exploration of health-seeking behavior emerges as a noteworthy aspect in our study. As individuals increasingly turn to the Internet for health information, the role of chatbot LLMs in shaping healthcare advice-seeking behavior becomes a compelling avenue for discussion. With the integration of technologies like Microsoft Bing (new Bing) with ChatGPT, new opportunities and risks surface in the realm of health information acquisition. Understanding how users interact with chatbot LLMs, the nature of information sought, and the potential impact on health decisions is pivotal. Opportunities may include enhanced accessibility to accurate information, empowering users in self-care. However, inherent risks, such as the potential for misinformation or misinterpretation, necessitate careful consideration. A thorough exploration of these dynamics will contribute valuable insights into evolving the healthcare advice-seeking behavior facilitated by chatbot LLMs.

### 4.2. Opportunities for Improvement and Advancement

In the ever-evolving AI-assisted healthcare conversations, identifying opportunities for improvement and advancement is crucial for addressing existing challenges and shaping the future trajectory of this dynamic field.

### 4.2.1. Enhancing Precision and Accuracy

Optimizing patient safety involves enhancing the precision and accuracy of AI models, especially LLMs, employed in medical chatbots [98]. Ongoing enhancements in these models are achievable through meticulous training on diverse and specialized medical datasets, effectively minimizing the potential risk of disseminating misinformation and bolstering the dependability of user responses. Implementing strategies such as fine-tuning LLMs for specific healthcare contexts and domains holds the potential to elevate their performance, ensuring a more nuanced understanding of user queries and, consequently, enhancing patient safety. Rigorous control measures, including continuous validation

and refinement, are paramount to mitigating risks and upholding the integrity of medical information disseminated by chatbots in healthcare settings.

### 4.2.2. Personalization and Context Awareness

There is significant potential in enhancing the personalization and context awareness of AI-assisted healthcare conversations [99]. Tailoring responses based on individual user profiles, medical histories, and preferences can create a more personalized and user-centric experience. Advancements in contextual understanding, incorporating factors such as patient context, emotional state, and real-time health data, can contribute to more nuanced and effective interactions, ultimately improving the overall quality of healthcare conversations.

### 4.2.3. Interdisciplinary Collaboration and Research

Opportunities for improvement extend to fostering interdisciplinary collaboration and research initiatives. Collaborations between AI researchers, healthcare professionals, ethicists, and legal experts can lead to comprehensive insights into the nuanced challenges of implementing AI in healthcare [100]. This collaborative approach can facilitate the development of robust frameworks, ethical guidelines, and innovative solutions that address the multifaceted aspects of AI-assisted healthcare conversations.

### 4.2.4. User Education and Engagement

Promoting user education and engagement represents another avenue for improvement. Initiatives focused on educating users about the capabilities and limitations of AI in healthcare conversations can enhance transparency and trust [101]. Encouraging active user participation in refining AI models, perhaps through feedback mechanisms, can contribute to the iterative improvement of medical chatbots, aligning them more closely with user expectations and needs. The improvement and future direction of AI-assisted healthcare conversations is summarized in Table 4.

**Table 4.** An overview of the key opportunities for improvement and advancement in the context of challenges and future directions in AI-assisted healthcare conversations.

| Opportunities for Improvement and Advancement | |
|---|---|
| Enhancing Precision and Accuracy | Continuous refinement of LLMs through targeted training on diverse medical datasets to reduce misinformation and improve reliability. |
| Personalization and Context Awareness | Tailoring responses based on user profiles, medical histories, and preferences for a more personalized and user-centric experience. |
| Interdisciplinary Collaboration and Research | Tailoring responses based on individual user profiles, medical histories, and preferences for a more personalized and user-centric experience. |
| User Education and Engagement | Initiatives to educate users about AI capabilities and limitations in healthcare, encouraging user feedback for iterative improvement. |

As we explore these opportunities for improvement and advancement, it becomes evident that the future of AI-assisted healthcare conversations is ripe with potential. By proactively addressing these opportunities, we can pave the way for more sophisticated, ethical, and user-centric applications, ultimately maximizing the positive impact of AI in the healthcare domain.

### 5. Conclusions

The conclusion highlights the dual nature of LLMs, like ChatGPT, acting as versatile tools across diverse fields while presenting potential drawbacks. Although LLMs enhance work efficiency, particularly in healthcare, concerns related to privacy and accuracy persist. NLP techniques, fundamental to LLMs, empower them to analyze diverse text sources, enabling symptom diagnosis and recommendations for patients. Despite the potential

advantages, ethical considerations, including privacy issues, and the need for specific training to enhance accuracy, remain pertinent. To navigate these challenges, concerted efforts are required to address privacy concerns, enhance public understanding, and regulate LLM usage. Future progress hinges on achieving consensus among governments and developers for effective regulation, with specific considerations for people with disabilities and elderly. Ethical concerns, including resource authenticity and potential plagiarism, underscore the need for continuous improvement and oversight. Achieving a balance between technological advancement and ethical considerations is crucial to foster trust and ensure the widespread, confident utilization of LLMs in healthcare conversations.

**Author Contributions:** Conceptualization, J.C.L.C. and K.L.; methodology, J.C.L.C., V.W. and K.L.; software, J.C.L.C., V.W. and K.L.; validation, J.C.L.C., V.W. and K.L.; formal analysis, J.C.L.C., V.W. and K.L.; investigation, J.C.L.C., V.W. and K.L.; resources, J.C.L.C. and K.L.; data curation, J.C.L.C., V.W. and K.L.; writing—original draft preparation, V.W. and J.C.L.C.; writing—review and editing, J.C.L.C. and K.L.; visualization, V.W. and J.C.L.C.; supervision, J.C.L.C.; project administration, J.C.L.C. and K.L.; funding acquisition, J.C.L.C. and K.L. All authors have read and agreed to the published version of the manuscript.

**Funding:** This work is supported by Planning and Dissemination Grants—Institute Community Support, Canadian Institutes of Health Research, Canada, under the grant numbers: CIHR PCS—168296 and CHIR PCS—191021.

**Institutional Review Board Statement:** Not applicable.

**Informed Consent Statement:** Not applicable.

**Data Availability Statement:** No new data were created.

**Acknowledgments:** The authors would like to thank the support from Leslie Sanders from the York University, Toronto, Canada. We acknowledge the use of ChatGPT [https://chat.openai.com/ accessed on 11 March 2024] to assist in proofreading the paper.

**Conflicts of Interest:** The authors declare no conflicts of interest.

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
