# Peer review of "Generative Pre-Trained Transformer-Empowered Healthcare Conversations: Current Trends, Challenges, and Future Directions in Large Language Model-Enabled Medical Chatbots"

_biomedinformatics, doi:10.3390/biomedinformatics4010047_

Round 1
Reviewer 1 Report
Comments and Suggestions for Authors
The paper gives a thorough review of AI-assisted healthcare chatbots. It covers current trends in this field as well as the challenges and risks we are currently facing. Considering the growing popularity of AI/LLM, I think this review paper can be very useful for researchers and developers who would like to integrate LLM into their own work. In addition, it can also promote and strengthen the collaboration between AI researchers and healthcare professionals.
The only suggestion I have is that I hope there is a list of the features that healthcare professionals would expect a chatbot to have. Knowing more about user expectations and their needs can be very helpful for AI researchers and developers.
Overall, this is a well-written and well-organized review paper. Therefore, I would recommend the paper for publication with minor revisions.
Reviewer 2 Report
Comments and Suggestions for Authors
This manuscript is fascinating because it deals with the implementation of AI in healthcare chatbots. The high number of references in the paper is commendable. That is also very easy to read.
1. I believe that the material and methods heading is incorrect. The material and methods in a scientific paper should explain the methodology used for the study, in this case, the review, for example, databases searched, keywords, etc. Alternatively, if it is a systematic review, the registry of the review in a database such as Prospero, etc. As this is not the case, the title of the heading should be eliminated and changed to another title, although the text of the section must remain as it is.
2. The above considerations can also be applied to the abstract.
3. The authors must explain the training of the models.
4. Authors should mention the concept of “hallucinations” in LLM , an important topic. The authors dealt with the topic but did not mention the concept(lines 324-332). There is abundant literature.
5. The consequence of this paragraph is “Because the LLMs are trained with the scientific literature that are copyrighted, the information or output like articles it is commanded to provide could be copied from those scientific journals and people might not know about this and causing plagiarism as a result “ could be that all LLM generated text, should be tested with a plagiarism detector.
6. One question that the authors should deal with is patient safety. What controls should be implemented to ameliorate risk for the patient?
7. Another interesting topic of this study is health-seeking behavior. This topic should be expanded. Some individuals use the Internet to seek health information. Will chatbot LLM be used in the same way. Now Microsoft bing can be used with ChatGPT. What opportunities and risk will that bring about?
8. In relation with the paragraph on “Adherence to existing healthcare regulations, such as HIPAA Health Insurance Portability and Accountability Act) in the United States,” the authors should expand the paragraph and explain how affect chat bot and IA.
9. In the same paragraph speak also about the The European Union’s Artificial Intelligence Act.
Reviewer 3 Report
Comments and Suggestions for Authors
The author extensively discusses GPT-authorized healthcare dialogues in the paper. I believe the strength of the discussion can be enhanced in the following areas:
The integration of conversational AI into existing healthcare systems might encounter technical challenges. Issues such as compatibility, interoperability with electronic health records, and the resistance of healthcare professionals to embrace new technologies could impede the smooth integration of AI into healthcare workflows.
Adherence to regulatory standards and healthcare laws is of utmost importance. The paper should delve into potential challenges related to ensuring that AI applications in healthcare align with existing regulations and standards, recognizing the regional variations in compliance requirements.
There is a risk of over-reliance on AI-driven solutions, potentially leading to reduced human oversight and decision-making. Striking a delicate balance between leveraging technology for efficiency and preserving the human touch in healthcare is crucial to guarantee that patients receive care that is both personalized and empathetic.
The implementation of advanced artificial intelligence systems, particularly large language models, can incur substantial financial implications. The costs associated with acquiring, implementing, and maintaining these technologies need careful consideration, especially for healthcare facilities operating with limited resources.
Reviewer 4 Report
Comments and Suggestions for Authors
This article is on GPT Empowered Healthcare Conversations. Regrettably, this article is not a comprehensive literature review. Both the method and structure deviate from established literature guidelines, disregarding the recommended PRISMA standards (http://prisma-statement.org/), which include adherence to the prescribed flow chart (http://www.prisma-statement.org/PRISMAStatement/FlowDiagram).
The title refers to current trends, challenges, and future directions, yet these elements are not thoroughly addressed. A full synthesis step is missing, as well as practical and theoretical implications.
Additionally, the keywords partial overlap and should be condensed.
Evidently, sections of the manuscript appear to be generated by artificial intelligence, irrespective of the manuscript's topic. The editorial board is encouraged to deliberate on whether such AI-generated content aligns with the journal's policies.
Furthermore, Table 1 could be better integrated into the text for improved coherence.
Concerning Figures 3 and 4, it is advisable not to include them in the discussion section, as it is unconventional to present figures in this part of the manuscript. Consider relocating them to a more suitable section according to standard practices.
The conclusion presents redundant information, while presenting only limited information for readers.
Comments on the Quality of English Language
Moderate editing of English language required
Reviewer 5 Report
Comments and Suggestions for Authors
Dear Authors,
I have read your manuscript titled "GPT Empowered Healthcare Conversations: Current Trends, Challenges, and Future Directions in LLM-Powered Medical Chatbots," and find it interesting. However, I believe the manuscript could benefit from expansion and improvement for better comprehension.
Please consider the following comments:
1. While there are numerous language models (LLMs) and LLM-based chatbots, the focus of this review is solely on ChatGPT, with no mention of other LLM-based medical chatbots. I recommend that the authors provide a broader perspective in this review, encompassing other relevant LLMs in the field.
2. In the introduction section, line 29 could be expanded to discuss the trajectory of artificial intelligence (AI) and machine learning adoption, particularly in the medical field. To provide a stronger background, the authors may consider citing the paper at https://www.mdpi.com/2079-9292/13/3/498, which delves into the adoption of AI in various medical specialties.
I believe implementing these recommendations will enhance the quality of the manuscript.
Best regards.
Comments on the Quality of English LanguageMinor editing of English language required
Round 2
Reviewer 1 Report
Comments and Suggestions for Authors
All my concerns have been addressed in the revised version of the paper.
Author Response
We would like to thank the insightful comments from the Reviewer again to improve the quality of this work.
Reviewer 2 Report
Comments and Suggestions for Authors
Thank you for incorporating all the comments; the manuscript has improved.
Author Response

(The authors gave the same response as above.)

Reviewer 3 Report
Comments and Suggestions for Authors
Accept in present form.
Author Response

(The authors gave the same response as above.)

Reviewer 4 Report
Comments and Suggestions for Authors
Unfortunately, the authors did not adress the most relevant shortcomings of their manuscript. The responses made in the cover letter are partly generic and not targeted to the actual request. Please thouroughly revise your article accodrding to publishing standards.
Comments on the Quality of English LanguageModerate editing of English language required
Author Response
We noted the comment on using AI to generate text in the manuscript. Although nowadays many authors would do that in preparing their manuscript, we did not do that in our work as we understood the limitation of using GPT to generate information (Chow et al Front. Artif. Intell. 2023;6:1166014). We investigated this issue and found that the reason may be due to we have used ChatGPT to help proofreading and editing some sections in the manuscript. Because of that, the text may look like generated by GPT. We followed the advice from the Editor to refer to the MDPI’s Updated Guidelines on Artificial Intelligence and Authorship (https://www.mdpi.com/about/announcements/5687), and made an acknowledgement of using ChatGPT in proofreading accordingly.
Reviewer 5 Report
Comments and Suggestions for Authors
Dear Authors,
All my comments are addressed.
Best.
Comments on the Quality of English LanguageMinor editing of English language required.
Author Response

(The authors gave the same response as above.)
